# Comparing the Carpal Tunnel Area and Carpal Boundaries in Patients with Carpal Tunnel Syndrome and Healthy Volunteers: A Magnetic Resonance Imaging Study

**DOI:** 10.3390/diagnostics15101205

**Published:** 2025-05-09

**Authors:** Yu-Ting Huang, Chii-Jen Chen, You-Wei Wang, Yi-Shiung Horng

**Affiliations:** 1Department of Physical Medicine and Rehabilitation, Taipei Tzuchi Hospital, Buddhist Tzuchi Medical Foundation, New Taipei City 23142, Taiwan; amandada0516@gmail.com; 2Department of Computer Science and Information Engineering, Tamkang University, New Taipei City 25137, Taiwan; cjchen@mail.tku.edu.tw; 3Department of Computer Science and Information Engineering, National Taiwan University, Taipei 10617, Taiwan; wei.tomato1112@gmail.com; 4Department of Medicine, Tzu Chi University, Hualien 97004, Taiwan

**Keywords:** carpal tunnel syndrome, magnetic resonance images, carpal tunnel area, median nerve

## Abstract

**Background:** Carpal tunnel syndrome (CTS) is a common neuropathy caused by compression of the median nerve (MN) within the carpal tunnel, which causes pain, paresthesia, or altered sensation. While a small carpal tunnel area is considered a risk factor for CTS, varying carpal tunnel dimensions in CTS patients have been obtained via axial computed tomography and magnetic resonance imaging (MRI). **Methods**: In this retrospective study, MR images from 49 CTS patients and 38 healthy controls were analyzed to investigate differences in the carpal tunnel area and carpal boundaries between the groups and to explore the relationships of these parameters with CTS severity. **Results**: Our findings revealed that compared with the controls, CTS patients presented significantly larger cross-sectional areas (CSAs) of the MN and carpal tunnel and increased MN flattening ratios. The CSAs of the MN showed moderate positive correlations with severity (r = 0.395, *p* < 0.001), symptom score (r = 0.354, *p* < 0.001), and disability score (r = 0.300, *p* < 0.001), while the carpal tunnel area showed weaker but significant correlations with severity (r = 0.268, *p* = 0.002), symptom score (r = 0.173, *p* = 0.026), and disability score (r = 0.183, *p* = 0.018). The ratios of the MN CSA to those of the carpal tunnel, the interior carpal boundary (ICB), the exterior carpal boundary (ECB), and the wrist were disproportionately greater in the CTS patients. Among them, both the MN-to-ICB and MN-to-ECB ratios had fair to good diagnostic values (area under the curve = 0.725 and 0.794, respectively). **Conclusions**: These results highlight the utility of MRI-derived CSA measurements and ratios in identifying pathophysiological changes in CTS patients, particularly crowding of the MN inside the carpal tunnel. Further studies are recommended to refine MRI-based diagnostic protocols for CTS.

## 1. Introduction

Carpal tunnel syndrome (CTS) is a neuropathy that involves entrapment of the median nerve (MN), occurring mostly at the inlet [1,2,3,4,5,6,7,8] or outlet [4,9,10,11,12] of the carpal tunnel. CTS manifests as pain, paresthesia, or altered sensation in the first three digits and the radial side of the ring finger and is often triggered by posture, movement, or even rest. The diagnosis of CTS relies on clinical symptoms; physical examinations, such as the Tinel sign and Phalen test; and electrophysiological studies [13].

Nerve conduction studies and electromyography are the gold standard methods for CTS diagnosis, but these tests are invasive and uncomfortable, with false-positive rates of 15% and false-negative rates of up to 10% [14,15]. Consequently, ultrasonography has gained popularity in recent decades because of its cost-effectiveness and accessibility [16,17,18,19,20,21,22,23]. Buchberger, et al. [24] first described four key sonographic features in CTS patients: (1) an increased cross-sectional area (CSA) of the MN at the pisiform and hamate bones, (2) a greater MN swelling ratio (ratio of the CSA of the MN at the pisiform level to that at the distal radius level), (3) an increased MN flattening ratio (ratio of the long axis of the MN to the short axis), and (4) significant palmar bowing of the flexor retinaculum. In addition, Kim, et al. [25] used ultrasonography and reported a significantly greater carpal tunnel CSA and nerve/tunnel index at the proximal level in affected hands than in the non-affected hands of patients with CTS. However, no significant difference was detected at the distal level. The authors also performed a subgroup analysis according to sex and reported significant differences in the CSA of the proximal carpal tunnel and the proximal nerve/tunnel index in female patients but not in male patients.

Carpal tunnel stenosis is believed to be a risk factor for CTS [26,27,28,29,30]. The CSA of the carpal tunnel is often used to evaluate the space available inside the tunnel for the MN, flexor tendons, and surrounding connective tissue [31,32,33,34]. Dekel and Coates [26,35] used axial computed tomography (CT) to measure the carpal tunnel area of female patients with CTS and reported that patients with CTS had significantly smaller tunnel areas than healthy subjects at both the proximal and distal levels. However, Winn et al. [36] reported significantly greater carpal tunnel areas in CTS patients than in healthy subjects at the level of the wrist crease and at 12 mm and 25 mm distal to the wrist crease.

Since conflicting results have been obtained via CT examinations, magnetic resonance imaging (MRI), with its superior resolution and more detailed soft tissue images, has also been used to evaluate the detailed anatomy of the carpal tunnel. Researchers have focused on measuring the CSA of the MN and carpal tunnel, the fractional anisotropy values on diffusion tensor imaging, the T2 signal intensity of the MN relative to the hypothenar muscle, and the three-dimensional volume of the carpal tunnel [37,38,39,40,41,42,43,44,45,46,47,48]. Gabra and Li [31] examined eight cadaveric hands, measured the cross-section of the distal carpal tunnel using MRI, and reported that the tunnel area comprised 18% of the CSA of the exterior carpal boundary (ECB).

Since the relationship between carpal tunnel size and CTS is still unclear, the purposes of this study were to (1) compare MR images of the carpal tunnel and its contents between patients with CTS and healthy volunteers, and (2) explore the relationship between carpal tunnel area and CTS severity.

## 2. Materials and Methods

### 2.1. Study Design

This retrospective study utilized clinical data and wrist MR images from 49 patients with CTS and 38 healthy controls previously collected in a National Science Council research project (NSC98-2314-B-303-008-MY2). The clinical data included basic demographic data, symptom scores, disability scores according to the Boston Carpal Tunnel Syndrome Questionnaire, and physical examinations and nerve conduction study results. All MRI scans were acquired using a 1.5T system (Signa Excite, GE Healthcare, Chicago, IL, USA) with the following imaging parameters: repetition time = 3200 ms, echo time = 95 ms, flip angle = 90°, field of view = 100 × 100 mm, slice thickness = 3.0 mm, inter-slice spacing = 3.3 mm, and matrix size = 256 × 192. Participants were positioned supine with their arms resting alongside the body and the wrists aligned in a neutral position, placed within a four-channel high-resolution wrist array coil. Although the conventional prone position with wrists extended overhead is considered optimal for scanner alignment, it is often uncomfortable to maintain. Therefore, to improve participant comfort and compliance—given a total scan time of approximately 40 min per subject—the supine position was chosen. The carpal tunnel, its osseous boundaries, and enclosed contents were measured at the pisiform level on axial T1-weighted MR images. The axial slice selected for analysis was defined as the level at which the pisiform bone exhibited the largest cross-sectional area, in accordance with established practices in CTS imaging research. As illustrated in the representative screenshot, image segmentation was performed manually by a trained musculoskeletal radiologist with over five years of experience, who was blinded to participants’ group allocation and clinical or electrophysiological information. Segmentation included the carpal tunnel, surrounding bony structures, and internal soft tissue contents. Area measurements were conducted using dedicated medical imaging software, with magnification and contrast settings optimized to ensure clear visualization and accurate delineation of nerves and tendons.

A researcher who was blinded to the study group and participant data manually delineated the boundaries of the MN and carpal tunnel. The carpal tunnel area was measured by tracing the margin of the carpal tunnel from the transverse carpal ligament on the volar side and palmar radiocarpal and ulnocarpal ligaments on the dorsal side, with the MN, the tendons of the flexor digitorum superficialis and profundus, and the flexor pollicis longus enclosed within. In addition to the MN and carpal tunnel area, the interior carpal boundary (ICB) was traced between the transverse carpal ligament and the interior border of the carpal bones (pisiform, triquetrum, lunate, and scaphoid), which includes the carpal tunnel area, tendons of the flexor carpal radialis, and surrounding connective tissues; the ECB, which includes the same structures as the ICB plus the carpal bones; and the wrist circumference, which was also manually outlined (Figure 1).

After delineating each target structure, including the MN, carpal tunnel area, ICB, ECB, and wrist joint, the researchers measured the CSAs, transverse axes, and anterior-posterior axes of the main structures, from which the flattening ratio of the MN (calculated as the transverse axis divided by the anterior-posterior axis), the aspect ratio of the wrist (calculated as the width-to-height ratio of the wrist joint), and the ratios of the CSAs of different regions were calculated.

CTS severity was scored according to electrophysiological findings as follows: 0, normal; 1, very mild, detected only by sensitive tests; 2, mild, slow sensory conduction velocity on finger/wrist measurement and normal terminal motor latency; 3, moderate, preserved sensory potential with motor slowing and distal motor latency of the abductor pollicis brevis < 6.5 ms; 4, severe, no sensory potential, preserved motor potential, and abductor pollicis brevis latency < 6.5 ms; 5, very severe, abductor pollicis brevis latency >6.5 ms; and 6, extremely severe, unrecordable sensory/motor potentials and abductor pollicis brevis amplitude < 0.2 mV [49].

### 2.2. Statistical Analysis

We performed descriptive statistics to summarize the participants’ basic demographic data, and we used Student’s t tests and the chi-squared test to compare the baseline demographic and clinical characteristics, including age, height, body mass index (BMI), and symptom and disability scores, educational level, and family income between the two study groups. Fisher’s exact tests were used to compare sex and dominant site between the two studied groups, since there were only five male participants in both groups and two left-handed CTS patients, and one left-handed healthy participant. In addition, considering the correlation between both hands of patients with bilateral CTS and healthy participants, we used generalized estimating equations to compare the physical examination results, nerve conduction study results, and MRI measurements between CTS patients and healthy controls after adjusting for age, sex, and BMI, ensuring that our results were valid and unbiased despite the dependent relationship between hands in individuals. Receiver operating characteristic curve analysis was performed to evaluate the diagnostic accuracy of several carpal measurement parameters using the area under the curve. *p* values less than 0.05 were considered to indicate statistical significance.

## 3. Results

Baseline demographic and clinical characteristics of 49 CTS patients and 38 healthy volunteers are summarized in Table 1. There were no significant differences between groups regarding age, sex, height, marital status, smoking habit, and hand dominance. CTS patients, however, had a higher BMI (25.8 ± 3.9 vs. 21.7 ± 3.0 kg/m^2^; *p* < 0.001), and markedly worse symptom (2.35 ± 0.77 vs. 1.07 ± 0.14) and disability scores (1.77 ± 0.69 vs. 1.01 ± 0.05; both *p* < 0.001). The employment status and education level were also significantly different between groups. Bilateral involvement was present in 84% of CTS cases.

Table 2 shows the clinical indicators. CTS patients presented significantly lower monofilament test scores; reduced grasp and palmar pinch power; and prolonged midpalm, distal sensory, and motor latencies of the MN. Wrist circumference was also greater in CTS patients than in the healthy controls. All the results were adjusted for age, sex, and BMI.

Table 3 presents the MRI measurements at the pisiform level. The CSA of the MN and the carpal tunnel area were significantly greater in the CTS patients than in the healthy controls. However, there were no significant differences in the CSAs of the ICB, ECB, or wrist. The MNs in the CTS patients had a greater transverse axis and flattening ratio.

Table 4 shows the ratios of various carpal tunnel measurements. The ratios of the MN CSA to the carpal tunnel area and the CSAs of the ICB, ECB, and wrist were all significantly greater in the CTS patients than in the healthy controls. Similar patterns were also observed for the ratios of the carpal tunnel area to the CSAs of ICB, ECB, and wrist, which were significantly greater in patients with CTS. No significant differences were observed in the ratio of the CSA of the ICB or ECB to the CSA of the wrist. Among these parameters, both the MN-to-ICB ratio and the MN-to-ECB ratio had fair to good diagnostic value (area under the curve = 0.725 and 0.794, respectively) (Figure 2).

Table 5 shows the results of the Pearson correlation analysis examining the relationships among CTS severity, symptom scores, disability scores, and various carpal tunnel measurements. The analysis revealed a strong positive correlation between CTS severity and the symptom score (r = 0.655, *p* < 0.01) and between the symptom score and the disability score (r = 0.844, *p* < 0.01). The CSA of the MN was moderately correlated with the severity, symptom score, and disability score. The flattening ratio of the MN and carpal tunnel area showed weaker but still significant correlations with the severity, symptom score, and disability score. Moderate to strong associations were found between the carpal tunnel area and the CSAs of the ICB and ECB.

## 4. Discussion

It is widely accepted that the CSA of the MN is greater in individuals with CTS than in healthy individuals, which reflects the mechanical compression and subsequent edema commonly associated with this CTS as a characteristic feature of this entrapment neuropathy. However, the relationship between carpal tunnel size and carpal tunnel content in CTS patients is still unclear. Our results indicate that compared with healthy individuals, CTS patients exhibit significant increases in the CSA of the MN. These findings can be regarded as a validation of the measurements performed in this study. We also found that the carpal tunnel area of CTS patients was significantly greater than that of healthy controls at the level of the pisiform bone. Since the carpal tunnel area was measured by tracing the transverse carpal ligament at the top, the carpal tunnel area increased when the transverse carpal ligament exhibited greater palmar bowing in patients with CTS, which has been described as one of the key features in patients with CTS [50]. Moreover, the volume of the carpal tunnel is influenced by tissue swelling, increased carpal tunnel pressure, and biomechanical factors such as wrist position and tendon loading. For example, increased pressure in the carpal tunnel affects the tunnel morphology, enlarging the CSA by 9.2% at 100 mmHg, and a significantly positive linear correlation between the palmar bowing of the transverse carpal ligament and the carpal tunnel area has been reported previously.

Although K. Monagle et al. [51] reported no significant difference in the carpal tunnel area between patients with CTS and healthy controls, we observed a significantly larger carpal tunnel area in patients with CTS, which is consistent with the findings of [22]. The discrepancies in these findings may stem from two main factors: (1) Variations in imaging modalities—some studies relied on CT [26,35,36,52], whereas others employed MRI [51,53,54]. Older investigations predominantly used CT, whereas more recent work often favors MRI. (2) Differences in measurement boundaries—some studies consider the “real” carpal tunnel (i.e., the region enclosing the MN and flexor tendons), whereas certain CT-based studies use the inner edges of the carpal bones, thereby including the flexor carpi radialis tendon and overlying soft tissue on the carpal bones. These additional structures can inflate the measured carpal tunnel area [31].

In addition to the above findings, we examined the correlation between carpal tunnel measurements and clinical indicators. Pearson’s correlation showed that CTS severity strongly correlated with symptom and disability scores, underscoring the link between severity, symptomatology, and functional impairment. The CSA of the MN and the flattening ratio had moderate correlations with severity and symptoms, whereas carpal tunnel area showed weaker associations. The carpal tunnel area/ICB ratio did not significantly correlate with severity. These observations highlight the importance of integrating clinical and anatomical assessments in CTS evaluation.

Changes in the synovial and intersynovial areas observed in patients with CTS have raised questions about the role of these changes in the development of CTS. Accordingly, it has been suggested that the CSA and volume of these connective tissues within the carpal tunnel may be key anatomical factors influencing CTS pathophysiology [35,36]. However, no statistically significant difference in these values has been demonstrated between CTS patients and controls. For example, S. Uchiyama et al. [55] reported that the CSAs of flexor tendons and the carpal tunnel area were greater in the CTS group than in the control group, but no significant differences in the synovial or intersynovial areas were detected between the groups. S. Tullie et al. [56] calculated the CSA of subsynovial connective tissue by subtracting the CSAs of flexor tendons and the MN from the carpal tunnel area. They reported no significant differences in the subsynovial connective tissue area between preoperative CTS patients and healthy controls, although paired comparisons of CTS patients before and after surgery revealed a significant increase in the CSA of the subsynovial connective tissue at the distal level.

Anthropometric characteristics have also been widely discussed in idiopathic CTS; however, the evidence regarding the relationship between anthropometric factors and CTS is still controversial. For example, small hands and square wrists have been reported to be associated with CTS [57,58,59,60,61,62], but other studies have failed to find a strong positive association between wrist circumference and CTS [63,64]. In our study, the aspect ratio (width-to-height ratio) of the wrist did not differ significantly between CTS patients and healthy controls. Several authors also underscore the interaction between elevated BMI and the wrist ratio [59,60,64]. Thiese et al. [60] noted a powerful effect modification by BMI, wherein the wrist ratio–CTS relationship was most pronounced in normal-weight individuals but partially masked among those with higher BMI. Conversely, Moghtaderi et al. [64] and Trybus et al. [59] identified high BMI itself as a risk factor, suggesting that obesity combined with a square wrist may further exacerbate CTS risk.

In addition, we examined the correlations between carpal tunnel measurements and clinical indicators, which revealed strong to moderate associations among CTS severity, symptom, and disability scores, the CSA, and the flattening ratio of the MN, whereas the carpal tunnel area presented weaker associations. These findings highlight the relevance of both clinical and anatomical assessments in evaluating CTS, with symptom and disability scores showing the strongest associations with CTS severity. We also found that the ratios of the MN CSA to the carpal tunnel area and the CSAs of the ICB, ECB, and wrist were significantly greater in CTS patients. Both the MN-to-ICB ratio and the MN-to-ECB ratio had fair to good diagnostic value (area under the curve = 0.725 and 0.794, respectively). This finding suggests that in CTS patients, the MN occupies a disproportionately larger area within the carpal tunnel. These findings highlight the potential of using MRI-derived CSA ratios as diagnostic markers, particularly in assessing the crowding of the MN within the carpal tunnel, which is a key pathophysiological feature of CTS. Moreover, similar to the swelling ratio and the median-to-ulnar ratio used in sonography [45], the use of the MN-to-ICB and MN-to-ECB ratios as diagnostic criteria can avoid the pitfalls associated with the use of a single cutoff value because the CSA of the MN might vary with population, age, or sex [36,37,65]. Although electrophysiological studies such as NCS and EMG remain the gold standard for diagnosing CTS, the use of image studies such as sonography or MRI provides additional diagnostic value in specific clinical scenarios. Several studies have demonstrated that MRI can reveal soft tissue abnormalities such as ganglion cysts, tenosynovitis, and bone cysts that may not be detected by electrophysiological studies, and which could influence surgical planning and outcomes [66]. Moreover, MRI proves particularly useful when electrophysiological results are inconclusive or when patients continue to experience symptoms despite negative electro-diagnostic findings [67,68]. MRI also serves as a valuable tool in postoperative assessment, helping to identify persistent median nerve compression or other complications that may explain unresolved symptoms [69]. Notably, MRI findings—such as increased signal intensity and nerve swelling—have been shown to correlate well with positive EMG results, further supporting its complementary role in CTS diagnosis [70]. Therefore, MRI should be considered not as a replacement, but as a supplemental imaging modality that can enhance diagnostic accuracy, especially in atypical, complex, or treatment-resistant cases.

Despite these insights, several limitations should be noted. First, although MRI offers valuable insights into the progression of CTS, the complexity of this technique may challenge less experienced clinicians or radiologists. Diagnostic errors range from 3% to 5%, and MRI errors are a significant contributor [71,72]. Measurement inaccuracies and parallax errors in imaging further complicate tunnel volume assessment [38,73]. L. R. Brigham et al. [71] analyzed study addenda and reported that communication errors were most common (44%), followed by insufficient history (21%), underreading (7%), and overreading (8%). The incidence of errors was 13.86/1000 on MR images and 12.45/1000 on CT images. J. P. Mogk and P. J. Keir [38] further evaluated the errors associated with off-axis imaging while measuring the carpal tunnel via three-dimensional reconstruction of MR images and reported that reorienting the imaging plane to align perpendicularly with the reconstructed carpal tunnel or external wrist angle revealed overestimations in the CSA and distortions in the carpal tunnel shape, which varied by wrist position.

Second, the relatively small sample size limits both the statistical power and the generalizability of our findings. A larger cohort would strengthen the conclusions and support more robust inferences. Third, the BMI of the two groups in our study differed significantly, introducing a potential confounding effect. To address this, we used generalized estimating equations adjusted for age, sex, and BMI. However, the CTS group had a lower employment rate and education levels than healthy participants; therefore, residual confounding cannot be entirely excluded.

Finally, the cross-sectional case–control design prevented us from determining whether smaller carpal tunnel dimensions serve as a risk factor for CTS or whether the increased carpal tunnel area in CTS patients is a consequence of elevated pressure, MN swelling, and flexor tendon changes leading to palmar bowing of the transverse carpal ligament. Prospective longitudinal studies are needed to clarify how the MN and carpal tunnel CSA evolve over time in patients with CTS and to elucidate the pathophysiological sequence of these anatomical changes.

## 5. Conclusions

This study revealed that the CSAs of both the MN and the carpal tunnel were greater in patients with CTS and had mild positive correlations with CTS severity and symptoms, and disability scores. In addition, both the MN-to-ICB and MN-to-ECB ratios had fair to good diagnostic value. These findings suggest that structural and proportional assessments of the carpal tunnel and MN provide valuable insights into CTS pathology, particularly the crowding of the MN inside the carpal tunnel. Further studies are recommended to refine MRI-based diagnostic protocols for CTS.

## Figures and Tables

**Figure 1 diagnostics-15-01205-f001:**
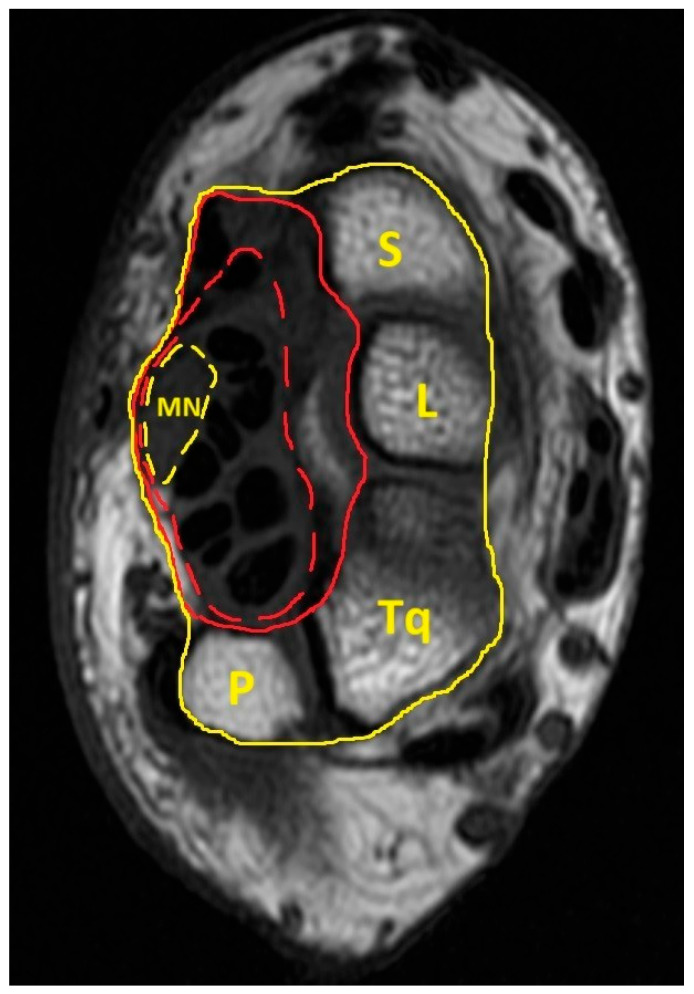
Transverse section of the proximal carpal tunnel. The anatomical structure encircled by the yellow dashed line is the median nerve; the anatomical structures delineated by the red dashed line are the components of the carpal tunnel (including the median nerve and tendons of the flexor pollicis longus, flexor digitorum superficialis, and flexor digitorum profundus). The anatomical structures delineated by the red solid line are the components of the interior carpal boundary (including the carpal tunnel, flexor carpi radialis, and surrounding connective tissue). The anatomical structures encircled by the yellow solid line are the components of the exterior carpal boundary (including the carpal tunnel, flexor carpi radialis, surrounding connective tissue, and carpal bones).

**Figure 2 diagnostics-15-01205-f002:**
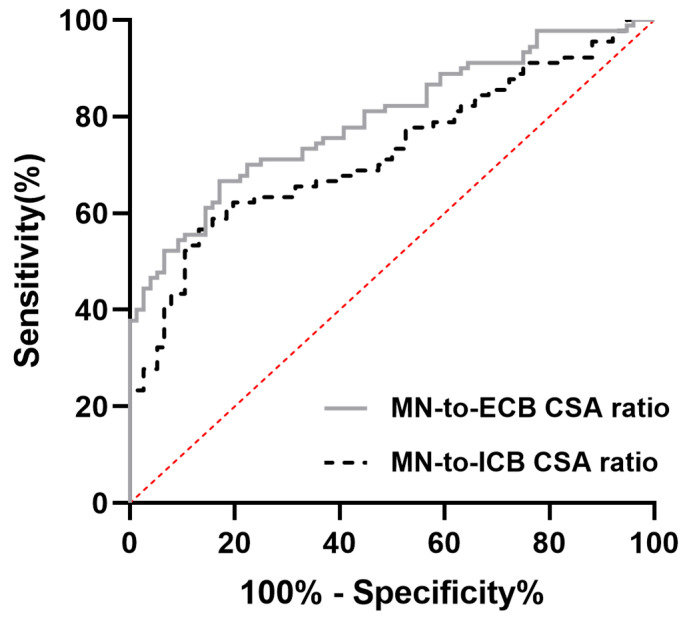
Receiver operating characteristic (ROC) curves comparing the diagnostic performance of the MN-to-ECB CSA ratio (solid gray line) and the MN-to-ICB CSA ratio (dashed black line) in differentiating patients from controls. The diagonal red line represents the line of no discrimination (AUC = 0.5).

**Table 1 diagnostics-15-01205-t001:** Baseline demographic and clinical characteristics of the participants.

Variables	Patient with CTS (*n* = 49)	Healthy Volunteer (*n* = 38)	*p* Value
Age, mean ± SD (years)	50.22 ± 9.93	48.39 ± 9.20	0.381
Gender, *n* (% of female)	44 (89.80%)	33 (86.84%)	0.742 ^a^
Height, mean ± SD (m)	157.38 ± 6.74	160.32 ± 7.86	0.064
BMI, mean ± SD (kg/m^2^)	25.78 ± 3.93	21.65 ± 2.96	<0.001
Symptom score, mean ± SD	2.35 ±0.77	1.07 ± 0.14	<0.001
Married, *n* (%)	36 (75.00%)	31 (81.58%)	0.465
Employed, *n* (%)	24 (48.98%)	29 (76.32%)	0.010
Smoking habit, *n* (%)	3 (6.12%)	1 (2.63%)	0.629 ^a^
Educational Level, *n* (%)			0.018
College/University or above	18 (36.73%)	25 (65.79%)	
Senior high school	19 (38.78%)	10 (26.32%)	
Junior high or less	12 (24.49%)	3 (7.89%)	
Disable score, mean ± SD	1.77 ± 0.69	1.01 ± 0.05	<0.001
Dominant side, *n* (% of right hand)	47 (95.92%)	37 (97.37%)	1.000 ^a^
Lesion side, *n* (% of right hand)			
Right-hand CTS	6 (12.24%)		
Left-hand CTS	2 (4.08%)		
Bilateral hand CTS	41 (83.67%)		

CTS, carpal tunnel syndrome. BMI, body mass index. *p*-value, comparison between CTS patients and healthy volunteers by Student’s *t* or ^a^ Fisher’s exact test.

**Table 2 diagnostics-15-01205-t002:** Comparison of clinical indicators between patients with CTS and healthy volunteers.

Variables	Patients with CTS (Patients/Hands = 49/90)	Healthy Volunteers (Persons/Hands = 38/76)	*p* Value
	Mean ± SD	Mean ± SD	
Monofilament test	30.063 ± 3.533	33.395 ± 1.980	<0.001
Grasp power (lb)	36.231 ± 14.314	50.623 ± 17.259	<0.001
Palmar pinch power (lb)	5.967 ± 3.378	8.261 ± 3.056	0.002
Lateral pinch power (lb)	8.790 ± 4.908	11.525 ± 3.836	0.006
Wrist circumference (cm)	15.598 ± 1.211	14.704 ± 1.078	<0.001
Midpalm latency (msec)	2.043 ± 0.509	1.505 ± 0.240	<0.001
Distal sensory latency of MN (ms)	3.581 ± 1.082	2.563 ± 0.299	<0.001
Distal motor latency of MN (ms)	4.931 ± 1.262	3.431 ± 0.273	<0.001
Amplitude of compound motor action potential of MN (mV)	12.955 ± 3.424	13.790 ± 3.585	0.276

CSA, cross-sectional area; MN, median nerve. *p*-value, comparison between CTS patients and healthy volunteers after adjusting for age, gender, and body mass index (generalized estimating equation).

**Table 3 diagnostics-15-01205-t003:** Outcomes of magnetic resonance image measurements at the pisiform level.

Variables	Patient with CTS (Patients/Hands = 49/90)	healthy Volunteer (Persons/Hands = 38/76)	*p* Value
	Mean ± SD (mm^2^)	Mean ± SD (mm^2^)	
CSA of MN	13.292 ± 3.076	10.411 ± 1.743	<0.001
CSA of CTA	171.101 ± 25.642	157.124 ± 26.679	0.034
CSA of ICB	279.327 ± 41.063	268.682 ± 42.234	0.733
CSA of ECB	758.254 ± 100.268	771.721 ± 111.959	0.826
CSA of wrist	1843.820 ± 238.826	1726.410 ± 258.983	0.470
Transverse axis of MN	6.453 ± 1.223	5.432 ± 0.816	<0.001
Anteroposterior axis of MN	2.704 ± 0.561	2.605 ± 0.331	0.732
Flattening ratio of MN	2.452 ± 0.532	2.113 ± 0.391	<0.001
Transverse axis of wrist	58.348 ± 4.231	57.121 ± 4.067	0.226
Anteroposterior axis of wrist	39.068 ± 3.009	37.498 ± 3.287	0.945
Aspect ratio of wrist	1.496 ± 0.087	1.527 ± 0.075	0.168

CSA, cross-sectional area; MN, median nerve. *p*-value, comparison between CTS patients and healthy volunteers after adjusting for age, gender, and body mass index (generalized estimating equation).

**Table 4 diagnostics-15-01205-t004:** Ratios of various carpal tunnel measurements.

Variables	Patient with CTS (Patients/Hands = 49/90)	Healthy Volunteer (Persons/Hands = 38/76)	*p* Value
CSA of MN/CSA of CTA	0.079 ± 0.020	0.067 ± 0.012	<0.001
CSA of MN/CSA of ICB	0.048 ± 0.012	0.039 ± 0.007	<0.001
CSA of MN/CSA of ECB	0.018 ± 0.004	0.014 ± 0.002	<0.001
CSA of MN/CSA of wrist	0.007 ± 0.002	0.006 ± 0.001	<0.001
CTA/CSA of wrist	0.094 ± 0.016	0.090 ± 0.017	0.009
CTA/CSA of ICB	0.615 ± 0.060	0.588 ± 0.068	0.005
CTA/CSA of ECB	0.227 ± 0.032	0.205 ± 0.028	0.005
CSA of ICB/CSA of wrist	0.153 ± 0.024	0.154 ± 0.027	0.253
CSA of ECB/CSA of wrist	0.414 ± 0.046	0.443 ± 0.065	0.550

CSA, cross-sectional area; MN, median nerve. *p*-value, comparison between CTS patients and healthy volunteers after adjusting for age, gender, and body mass index (generalized estimating equation, GEE).

**Table 5 diagnostics-15-01205-t005:** Correlations between various carpal tunnel measurements and clinical indicators.

Pearson Correlation	Severity	Symptom Score	Disable Score	CSA of MN	Flattening Ratio of MN	CTA	CSA of ICB	CSA of ECB
Severity	1.000	0.655 **	0.533 **	0.395 **	0.268 **	0.212 **	0.160 *	−0.005
Symptom Score	0.655 **	1.000	0.844 **	0.354 **	0.310 **	0.173 *	0.098	−0.031
Disable score	0.533 **	0.844 **	1.000	0.300 **	0.331 **	0.183 *	0.120	0.015
CSA of MN	0.395 **	0.354 **	0.300 **	1.000	0.103	0.304 **	0.223 **	0.104
Flattening ratio of MN	0.268 **	0.310 **	0.331 **	0.103	1.000	0.155 *	0.058	−0.009
CTA	0.212 **	0.173 *	0.183 *	0.304 **	0.155 *	1.000	0.768 **	0.488 **
CSA of ICB	0.160 *	0.098	0.120	0.223 **	0.058	0.768 **	1.000	0.585 **
CSA of ECB	−0.005	−0.031	0.015	0.104	−0.009	0.488 **	0.585 **	1.000

**: correlation is significant at the 0.01 level (2-tailed); *: correlation is significant at the 0.05 level (2-tailed).

## Data Availability

The data analyzed in this original article are included in this manuscript. Further datasets used and/or analyzed in this study are available from the corresponding author on request.

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
