# Peer review of "Comparing the Carpal Tunnel Area and Carpal Boundaries in Patients with Carpal Tunnel Syndrome and Healthy Volunteers: A Magnetic Resonance Imaging Study"

_diagnostics, 2025, doi:10.3390/diagnostics15101205_

Round 1

Reviewer 1 Report

Comments and Suggestions for Authors

The manuscript has a well described design.

the introduction needs improvement by adding the correct references especially on line 33.

methods: there is no patient background as social occupation, medical history, any additional pharmaceutical drugs which are the most important for anyone who has CTS. the paragraphs are written chaotically and there is no flow in the presentation. The MRI and CT have some protocols for their performance. Let alone why patients underwent CT scan for CTS with so much radiation? No one described the technique and equipment of the imaging studies.

the results support the given information and if you revise the paper, they should be updated.

Discussion: the discussion supports your findings, however, there is no comment of why one patient should perform so expensive imaging studies when EMG is pretty good in diagnosing CTS .

Author Response

Comment 1: the introduction needs improvement by adding the correct references especially on line 33.

Response 1:

We thank the reviewer for highlighting the need to strengthen the Introduction with appropriate references. In response, we have revised line 33 to include a relevant and recent citation that supports the diagnostic criteria for carpal tunnel syndrome (CTS), specifically regarding the use of clinical symptoms, physical examination (Tinel’s and Phalen’s tests), and electrophysiological studies.

The revised sentence now reads: "The diagnosis of CTS relies on clinical symptoms; physical examinations, such as the Tinel sign and Phalen test; and electrophysiological studies [1](Saggar et al., 2024)."

The cited study by Saggar et al. (2024) provides a comprehensive analysis of neurophysiological severity grading in CTS and correlates it with clinical signs, thereby substantiating the statement. The modification has been implemented in the revised manuscript (Introduction section, page 1; line 28-32).

We appreciate the reviewer’s suggestion, which helped improve the clarity and accuracy of our introductory discussion.

Comment 2: methods: there is no patient background as social occupation, medical history, any additional pharmaceutical drugs which are the most important for anyone who has CTS. the paragraphs are written chaotically and there is no flow in the presentation. The MRI and CT have some protocols for their performance. Let alone why patients underwent CT scan for CTS with so much radiation? No one described the technique and equipment of the imaging studies. The results support the given information and if you revise the paper, they should be updated.

Response 2:

We thank the reviewer for the detailed and valuable feedback.

  1. As this was a retrospective study, complete baseline demographic data, including occupation, medical history, and any additional pharmaceutical drugs use, were not consistently available for all participants. However, in response to your comment, we have updated the Result section (page 3-4; line 117-119) and Table 1 (page 4-5; line 148-150) to include the available patients’ background information including We appreciate the reviewer’s insightful comment. In response, we have updated Table 1 to include additional patient background information, including marital status, employment status, smoking habits, and educational level from our records to the extent possible.
  2. We would like to clarify that no CT scans were performed in this study. All imaging analyses were based on MRI, which is a non-radiative modality. We apologize for any confusion and have revised the text accordingly to ensure this distinction is clear.
  3. Additionally, we have improved the organization and clarity of the Methods section (page 2-3; line 65-80)_to enhance the flow and logical presentation. Detailed MRI parameters, imaging protocols, and equipment specifications have now been added to ensure transparency and reproducibility.

We appreciate your comments, which have helped us significantly improve the structure and clarity of the manuscript.

Comment 3: Discussion: the discussion supports your findings, however, there is no comment of why one patient should perform so expensive imaging studies when EMG is pretty good in diagnosing CTS.

Response 3:

We appreciate the reviewer’s insightful comment regarding the role of MRI in CTS diagnosis. While we acknowledge that electrophysiological studies, such as nerve conduction study (NCS) or EMG is widely regarded as the gold standard for diagnosing CTS due to its ability to detect functional nerve impairments, but these tests are uncomfortable and even invasive, with false-positive rates around 15% and false-negative rates of up to 10% [2,3]. Therefore, imaging studies, such as sonography and MRI, also play a valuable and complementary role in selecting clinical contexts, especially for MRI which has better resolutions than sonography.

First, MRI offers the advantage of directly visualizing structural abnormalities that may underlie or contribute to CTS, such as ganglion cysts, tenosynovitis, or degenerative bone lesions. Identification of these pathologies can significantly influence surgical decision-making and improve clinical outcomes [4].

Second, in cases where NCS results are equivocal or normal despite persistent clinical symptoms, MRI can serve as a valuable adjunct by providing detailed anatomical information that supports or refines the diagnosis [5,6].

Third, MRI is particularly helpful in the postoperative setting, as it allows for objective assessment of the median nerve and surrounding tissues to determine whether residual or recurrent compression is contributing to unresolved symptoms [7].

Finally, multiple studies have demonstrated a correlation between NCS-confirmed CTS and abnormal MRI findings, such as increased signal intensity or nerve swelling, further validating MRI as a supportive diagnostic modality [8].

In conclusion, we respectfully propose that MRI should not be seen as a replacement for EMG, but rather as a complementary imaging tool that can enhance diagnostic confidence in situations where:

  • NCS/EMG results are inconclusive;
  • detailed anatomical clarification is required;
  • additional soft tissue pathology is suspected; or
  • postoperative evaluation is necessary.

We have included this clarification in the revised manuscript (Discussion section, page 8, lines 224-235), and we thank the reviewer for the opportunity to expand on this important point.

  1. Saggar, S.K.; Thaman, R.G.; Mohan, G.; Kumar, D. Mapping Neurophysiological Patterns in Carpal Tunnel Syndrome: Correlations With Tinel's and Phalen's Signs. Cureus 2024, 16, e58168, doi:10.7759/cureus.58168.
  2. Padua, L.; Padua, R.; Aprile, I.; D'Amico, P.; Tonali, P. Carpal tunnel syndrome: relationship between clinical and patient-oriented assessment. Clin Orthop Relat Res 2002, 128-134, doi:10.1097/00003086-200202000-00013.
  3. Okura, T.; Sekimoto, T.; Matsuoka, T.; Fukuda, H.; Hamada, H.; Tajima, T.; Chosa, E. Efficacy of Diagnosing Carpal Tunnel Syndrome Using the Median Nerve Stenosis Rate Measured on Ultrasonographic Sagittal Imagery: Clinical Case-Control Study. Hand (N Y) 2023, 18, 133s-138s, doi:10.1177/15589447211017225.
  4. Onen, M.R.; Kayalar, A.E.; Ilbas, E.N.; Gokcan, R.; Gulec, I.; Naderi, S. The Role of Wrist Magnetic Resonance Imaging in the Differential Diagnosis of the Carpal Tunnel Syndrome. Turk Neurosurg 2015, 25, 701-706, doi:10.5137/1019-5149.Jtn.10754-14.2.
  5. Lacotte, B.; Pierre-Jérôme, C.; Coessens, B.; Shahabpour, M.; Durdu, J. [Carpal tunnel syndrome. Comparative studies of pre- and postoperative magnetic resonance and electromyography]. Ann Chir Main Memb Super 1991, 10, 300-307.
  6. MusluoÄŸlu, L.; Celik, M.; Tabak, H.; Forta, H. Clinical, electrophysiological and magnetic resonance imaging findings in carpal tunnel syndrome. Electromyogr Clin Neurophysiol 2004, 44, 161-165.
  7. Crnković, T.; Trkulja, V.; Bilić, R.; Gašpar, D.; Kolundžić, R. Carpal tunnel and median nerve volume changes after tunnel release in patients with the carpal tunnel syndrome: a magnetic resonance imaging (MRI) study. Int Orthop 2016, 40, 981-987, doi:10.1007/s00264-015-3052-8.
  8. Göktürk, Åž.; Göktürk, Y.; Koç, A.; Payas, A. Comparison of median nerve area measurement between MRI and electromyography in patients diagnosed with carpal tunnel syndrome. Adv Clin Exp Med 2024, doi:10.17219/acem/187054.

Reviewer 2 Report

Comments and Suggestions for Authors

Authors present a retrospective study on CTS. They use MR imaging to this end and find correlations between different biomarkers and CTS severity. In general English, figures and structure of the document are correct. The methods section could be better or more thoroughly explained but is understandable. The discussion section is quite large and covers almost all points found in this paper. Introduction, as mentioned below, would require the use of newer references. As it is presented here, every reference there is old and does not allow to present a current state of the art. Needs changes made in it.

Major changes

In abstract maybe some values could be introduced instead of terms like mildly correlated.

I would add the last paragraph of introduction in which ultrasounds were used (54 to 58 in page 2) to the ultrasound section of the introduction (finished line 40).

Introduction is logical and well presented. It justifies the study. One detail that is problematic though is that references from this section in general are something like 15 to 30 years old. Newer references are presented in the discussion. So do not know if there is a possibility of incorporating newer works to the introduction to make the sate of the art more actual.

In the methods section. More information on MRI  parameters should be given. More information on the researcher that performed image segmentation and analysis should be provided. How did he/she do the work? All this to make the study reproducible. Also in statistics section indication of parametricity of data should also be commented.

 In the results section. Authors have to clarify from table 2 to 5 the patients to hand ratio I.e. in patients with CTS 49/90. Where did we lose 8 hands here? Did all patients have CT in both hands or not?  

Minor changes

Join two paragraphs of abstract.

0.742a What does this a mean in table 1?

Author Response

Comment 1: The methods section could be better or more thoroughly explained but is understandable. The discussion section is quite large and covers almost all points found in this paper. Introduction, as mentioned below, would require the use of newer references. As it is presented here, every reference there is old and does not allow to present a current state of the art. Needs changes made in it.

Response 1:

We sincerely thank the reviewer for the constructive feedback. In response, we have revised the Methods section to enhance clarity and ensure that all procedures are more thoroughly described for better reproducibility.

Regarding the Introduction, we fully acknowledge the reviewer’s concern about outdated references. We have accordingly updated the section by incorporating several more recent and relevant references to better reflect the current state of the art in carpal tunnel syndrome research. These changes aim to strengthen the scientific foundation and relevance of the study background.

We appreciate the reviewer’s insightful comments, which have helped improve the overall quality of the manuscript.

Comment 2: In abstract maybe some values could be introduced instead of terms like mildly correlated.

Response 2:

We appreciate the reviewer’s suggestion to provide greater clarity and specificity regarding the correlation between anatomical measurements and clinical severity in CTS. In response, we have revised the original sentence:

"The CSAs of both the MN and the carpal tunnel were mildly positively correlated with CTS severity and with symptom and disability scores."

to a more detailed version that includes correlation coefficients and statistical significance:

"The CSAs of the MN showed moderate positive correlations with severity (r = 0.395, p < 0.001), symptom score (r = 0.354, p < 0.001), and disability score (r = 0.300, p < 0.001), while the carpal tunnel area showed weaker but significant correlations with severity (r = 0.268, p = 0.002), symptom score (r = 0.173, p = 0.026), and disability score (r = 0.183, p = 0.018)."

This revision has been made to enhance the precision of our findings and provide readers with a clearer interpretation of the data. The updated text appears in the Abstract section (page 1, line 17-20) of the revised manuscript.

Comment 3: I would add the last paragraph of introduction in which ultrasounds were used (54 to 58 in page 2) to the ultrasound section of the introduction (finished line 40).

Response 3:

Thank you for your helpful suggestion. As advised, we have moved the last paragraph of the introduction (lines 54–58 on page 2), which discusses the use of ultrasound, to the earlier ultrasound section of the introduction (after line 40). This restructuring improves the logical flow and thematic coherence of the introduction.

The revision has been implemented in the updated manuscript accordingly. (page 2; line 39-43)

Comment 4: Introduction is logical and well presented. It justifies the study. One detail that is problematic though is that references from this section in general are something like 15 to 30 years old. Newer references are presented in the discussion. So do not know if there is a possibility of incorporating newer works to the introduction to make the state of the art more actual.

Response 4:

Thank you for your thoughtful feedback. In response to your comment, we have updated the Introduction section by incorporating several more recent references to better reflect the current state of the art and ensure the literature review is up to date. These additions enhance the relevance and timeliness of the background information presented in the manuscript.

Comment 5: In the methods section. More information on MRI parameters should be given. More information on the researcher that performed image segmentation and analysis should be provided. How did he/she do the work? All this to make the study reproducible. Also in statistics section indication of parametricity of data should also be commented.

Response 5:

We appreciate the reviewer’s valuable feedback regarding the need for greater methodological transparency.

We have revised the Methods section to include detailed MRI acquisition parameters and clarified the process of image segmentation and analysis to enhance the reproducibility of our study. The updated paragraph now reads as follows (Methods section; page 2-3; line 65-80)): “All MRI scans were acquired using a 1.5T system (Signa Excite, GE Healthcare) with the following imaging parameters: repetition time= 3200 ms, echo time= 95 ms, flip angle = 90°, field of view= 100 × 100 mm, slice thickness = 3.0 mm, inter-slice spacing = 3.3 mm, and matrix size = 256 × 192. Participants were positioned supine with their arms resting alongside the body and the wrists aligned in a neutral position, placed within a four-channel high-resolution wrist array coil. Although the conventional prone position with wrists extended overhead is considered optimal for scanner alignment, it is often uncomfortable to maintain. Therefore, to improve participant comfort and compliance—given a total scan time of approximately 40 minutes per subject—the supine position was chosen. The carpal tunnel, its osseous boundaries, and enclosed contents were measured at the pisiform level on axial T1-weighted MR images. The axial slice selected for analysis was defined as the level at which the pisiform bone exhibited the largest cross-sectional area, in accordance with established practices in CTS imaging research. As illustrated in the representative screenshot, image segmentation was performed manually by a medical doctor who was blinded to participants’ group allocation and clinical or electrophysiological information. Segmentation included the carpal tunnel, surrounding bony structures, and internal soft tissue contents. Area measurements were conducted using dedicated medical imaging software, with magnification and contrast settings optimized to ensure clear visualization and accurate delineation of nerves and tendons.”

Also, we have revised the Statistics section to explicitly address the parametricity of the data and clarify the statistical tests used. The updated paragraph now reads as follows (page 3; line 101-112): "We performed descriptive statistics to summarize the participants’ basic demographic data, and we used Student’s t tests and the chi-squared test to compare the baseline demographic and clinical characteristics, including age, height, body mass index (BMI), and symptom and disability scores, educational level and family income between the two study groups. Fisher’s exact tests were used to compare sex and dominant site between two studied groups since there were only five male participants in both group and two left-handed CTS patients and one left-handed healthy participant. In addition, considering the correlation between both hands of patients with bilateral CTS and healthy participants, we used generalized estimating equations to compare the physical examination results, nerve conduction study results, and MRI measurements between CTS patients and healthy controls after adjusting for age, sex and BMI, ensuring that our results were valid and unbiased despite the dependent relationship between hands in individuals. Receiver operating characteristic curve analysis was performed to evaluate the diagnostic accuracy of several carpal measurement parameters using the area under the curve. P values less than 0.05 were considered to indicate statistical significance"

Comment 6: In the results section. Authors have to clarify from table 2 to 5 the patients to hand ratio I.e. in patients with CTS 49/90. Where did we lose 8 hands here? Did all patients have CT in both hands or not?  

Response 6:

Thank you for pointing this out. As shown in Table 1, a total of 49 patients with CTS contributed 90 hands because there were 8 patients with unilateral CTS and 41 patients with bilateral involvements. Therefore, the unaffected hands of 8 unilateral CTS patients were not included in the statistical analysis. Therefore, the patients to hand ratio in patients with CTS shown in Table 2 to 5 was 49/90.

Comment 7: Join two paragraphs of abstract. 0.742a What does this a mean in table 1?

Response 7:

Thank you for your comment. We have revised the abstract by combining the two paragraphs as suggested. Regarding Table 1, the superscript “a” refers to the use of Fisher’s exact test, and we have now clarified this in the table legend to avoid any confusion.

Round 2

Reviewer 1 Report

Comments and Suggestions for Authors

The manuscript has been improved by the authors. The scientific soundness is improved, the methodology has been improved.